# Nucleosome landscape reflects phenotypic differences in *Trypanosoma cruzi* life forms

**Alex R. J. Lima**[1,2]**, Christiane B. de Araujo**[1,2]**, Saloe Bispo**[1,2]**, José Patané**[1,2]**, Ariel M. Silber**[3]**, M. Carolina Elias**[1,2]*****, Julia P. C. da Cunha**[1,2]*****

**1** Laboratório de Ciclo Celular, Instituto Butantan, São Paulo, Brazil, **2** Center of Toxins, Immune Response and Cell Signaling (CeTICS), Instituto Butantan, São Paulo, Brazil, **3** Laboratory of Biochemistry of Tryps–LaBTryps, Department of Parasitology, Institute of Biomedical Sciences, Universidade de São Paulo, São Paulo, Brazil

* carolina.eliassabbaga@butantan.gov.br (MCE); julia.cunha@butantan.gov.br (JPCC)

## Abstract

*Trypanosoma cruzi* alternates between replicative and nonreplicative life forms, accompanied by a shift in global transcription levels and by changes in the nuclear architecture, the chromatin proteome and histone posttranslational modifications. To gain further insights into the epigenetic regulation that accompanies life form changes, we performed genome-wide high-resolution nucleosome mapping using two *T. cruzi* life forms (epimastigotes and cellular trypomastigotes). By combining a powerful pipeline that allowed us to faithfully compare nucleosome positioning and occupancy, more than 125 thousand nucleosomes were mapped, and approximately 20% of them differed between replicative and nonreplicative forms. The nonreplicative forms have less dynamic nucleosomes, possibly reflecting their lower global transcription levels and DNA replication arrest. However, dynamic nucleosomes are enriched at nonreplicative regulatory transcription initiation regions and at multigenic family members, which are associated with infective-stage and virulence factors. Strikingly, dynamic nucleosome regions are associated with GO terms related to nuclear division, translation, gene regulation and metabolism and, notably, associated with transcripts with different expression levels among life forms. Finally, the nucleosome landscape reflects the steady-state transcription expression: more abundant genes have a more deeply nucleosome-depleted region at putative 5' splice sites, likely associated with trans-splicing efficiency. Taken together, our results indicate that chromatin architecture, defined primarily by nucleosome positioning and occupancy, reflects the phenotypic differences found among *T. cruzi* life forms despite the lack of a canonical transcriptional control context.

## Author summary

Trypanosomes have profound changes in their morphology and gene expression along life forms, with clear changes on expression of virulence factors, on transcriptional activity and on proliferation capacity. How all these processes are achieved remains unsolved especially considering that, in these organisms, gene regulation relies mainly on

**Data Availability Statement:** The datasets generated and/or analyzed during the current study are available in the NCBI BioProject repository, under the accession code PRJNA665060 and

BioSample codes SAMN16241276, SAMN16241277, SAMN16241279, SAMN16241280, SAMN16241281 and SAMN16241282.

**Funding:** This work was supported by fellowships from the Sao Paulo Research Foundation (FAPESP) by grants 18/15553-9 (JPCC), 13/07467-1 (JPCC,MCE), 19/19690-3 (ARJL), 18/14432-3 (AMS,MCE, JPCC), 16/50050-2 (MCE) and by the Serrapilheira Institute (grant number Serra-1709-16865 to JPCC). MCE and AMS have a fellowship from the National Council for Scientific and Technological Development (CNPq). The funders had no role in study design, data collection and analysis, decision to publish, or preparation of the manuscript.

**Competing interests:** The authors have declared that no competing interests exist.

posttranscriptional mechanisms. Here, we show that chromatin organization, held primarily on nucleosome positioning and occupancy, differs at strategic genomic regions and reflects phenotypic differences observed among *T. cruzi* life forms. The putative transcription starts sites and multigenic family members -that code for virulence factors, are differentially enriched on dynamic nucleosomes among replicative and nonreplicative *T. cruzi* forms. In addition to that, genes associated with DNA replication, cytokinesis, transcription and translation regulation are mainly enriched on dynamic nucleosomes.

## Introduction

Chromatin is the template of essential cellular processes such as transcription, replication and repair, and therefore, its structure and organization must be finely regulated. Nucleosomes are the basic unit of chromatin and are composed of 147 bp of DNA wrapped around an octamer of histones. Nucleosome positioning (where nucleosomes are located with respect to the genomic DNA sequence) and occupancy (local nucleosome density in a cell population) are important features of chromatin organization and influence crucial aspects of epigenetic regulation [1]. Their positioning is governed both by intrinsic DNA sequences and by trans-factors such as ATP-dependent remodelers, transcription factors and the RNA Polymerase elongation [2,3]. One technique widely used to map nucleosome positioning and occupancy uses a *Micrococcus* nuclease enzyme to digest DNA not bound to nucleosomes, followed by deep sequencing (MNase-seq)[4].

Nucleosomes are extremely dynamic, and their organization and composition are crucial for gene regulation and DNA replication as they regulate DNA sequence accessibility to regulatory complexes [5]. Many efforts have been employed to establish a relationship between chromatin organization into nucleosomes and RNA metabolism. In yeast, a unique nucleosome architecture around the transcription start sites (TSSs) is found, comprising a nucleosome-depleted region (NDR) just upstream of the TSS, followed by at least three well-positioned nucleosomes downstream [6]. In metazoan and plant genomes, nucleosomes with well-defined positions were also found around TSSs and associated with splicing sites, but most nucleosomes have a fuzzy pattern[4]. Promoter regions are depleted of nucleosomes, and experiments in yeast using conditional RNA Pol subunit mutants indicate that transcriptional activity may influence the width of the NDR at a TSS as well as promote nucleosome sliding [7]. Therefore, RNA polymerase influences nucleosome landscape dynamics, generating a thermodynamically unfavorable structure. Recently, single-cell analysis of MNase-seq data has started to clarify the association of nucleosome occupancy and transcription levels: uniformly spaced but poorly positioned nucleosomes were found at silent genes, whereas well-positioned but irregularly spaced nucleosomes were found at active genes [8].

Trypanosomatid protists, such as *Trypanosoma cruzi*, *Trypanosoma brucei*, and *Leishmania sp*., include the causative agents of important human diseases [9]. Two important biological features make trypanosomes interesting cell models for epigenetic studies. The first is the non-canonical gene regulation presented by these organisms [10]. They have a peculiar genomic structure and regulation: protein-coding genes are organized into large, directional gene clusters (or polycistronic transcription units, PTUs) that can range from ~30 to 500 kb on a genome-wide scale. Additionally, PTUs are flanked by divergent or convergent strand-switch regions (dSSRs and cSSRs, respectively) that function as TSSs and TTSs (transcription initiation and termination sites, respectively) [11,12]. Trypanosome mRNAs are also subjected to a trans-splicing mechanism, as their mRNA receives at their 5' end a spliced leader (SL)

sequence, which originates from another transcript, concomitantly with the polyadenylation of the upstream transcript [13]. Therefore, gene expression in trypanosomes is remarkable, as transcription is polycistronic, which means that RNA polymerase transcribes long transcripts containing several open reading frames (ORFs), resulting in a lack of gene-specific regulation at transcription initiation; rather, gene expression control relies mainly on posttranscriptional mechanisms [10,14].

Another interesting biological feature of trypanosomes is their capacity to adapt to drastic environmental changes triggered by changes in pH, nutritional availability, adhesion status and temperature shifts to complete their digenetic life cycle. These parasites alternate between life forms with clear phenotypic changes in cell morphology, replication and infective capacity. *T. cruzi* epimastigote forms are noninfective and replicative forms that live in the midgut of triatomine insects, while tissue cultured derived trypomastigote (TCT) forms are infective and nondividing forms that live inside the mammalian host. In addition to these features, epimastigotes have a much higher global transcription rate than TCT forms [15]. Likewise, in *T. brucei* stumpy and metacyclic (both non-replicative) forms have almost no incorporation of 3H uridine into nascent mRNAs and 35S cysteine incorporation into newly translated protein, respectively, and both have very low levels of mRNAs [16,17]. In addition, it is also impossible to induce expression from an RNA pol I promoter in the stumpy form[18].

Similar to other eukaryotes, trypanosomatids organize their DNA in nucleosomes; however, neither 30 nm fibers nor chromosome condensation is observed during mitosis [19], and their histone primary sequence is variable compared to that of other eukaryotes [20]. In. *T. cruzi*, histone posttranslational modifications (PTMs) have been identified by us and others, and their association with the cell cycle, DNA replication, DNA damage and gene expression has been addressed [21–25]. Epigenetic mechanisms such as deposition of histone variants, histone PTMs and base J (a β-D-glucopyranosyloxymethyluracil found in DNA from kinetoplastids) were described at trypanosomatid dSSRs and cSSRs [26–29], most likely influencing both local chromatin structure and function as an attractive platform for related regulatory complexes.

Nucleosome organization has already been mapped in *T. brucei* and Leishmania spp. [30–32] by MNase-seq. In *T. brucei*, the RNA polymerase II initiation regions do not show the typical nucleosome landscape observed in other eukaryotes but clearly indicate the presence of a well-positioned nucleosome at the splice acceptor site within each CDS of the PTU [31,32]. In addition, nucleosomes are depleted from the locus of variable surface protein (VSG), a virulence factor [33], and only subtle differences in nucleosome position were observed among the procyclic and bloodstream forms of *T. brucei* [31]. However, in addition to their obvious differences in environmental growth conditions, both life forms are replicative. Here, for the first time, we performed a genome-wide analysis of nucleosome positioning in two *T. cruzi* life forms by MNase-seq and integrated these results with published transcriptomic datasets. We observed that *T. cruzi* nucleosome positioning and occupancy at strategic genomic regions differ among life cycle forms, likely associated with some critical phenotypic differences. We discuss these findings considering the unique gene regulation scenario found in trypanosomatids.

## Results

### Nucleosome distribution along the *T. cruzi* genome

To investigate the possible impact of epigenetic changes based on nucleosome positioning and occupancy (herein defined as NPO) on the phenotypic differences observed among life forms, we performed MNase-seq in two life forms of this parasite. We used the CL-Brener strain, a hybrid strain with two different haplotypes ("Esmeraldo-like" and "Non-Esmeraldo-like"), whose genome was previously sequenced [11] and mapped into 41 *in silico* chromosomes [34].

In short, chromatin from epimastigotes and TCTs (in biological triplicates) was digested with the *micrococcal nuclease* enzyme, which preferentially cleaves the linker DNA, leaving mono-nucleosomes (S1A Fig). The mononucleosome band (approximately 150 bp) was gel-purified and paired-end sequenced at high depth using an Illumina NextSeq 500. To simplify our analysis, reads were mapped only to the CL-Brener Esmeraldo-like haplotype (S1B Fig); thus, of the total ~21% to 15% of reads that were not mapped across the replicates (S1C Fig), some may be specific to the non-Esmeraldo-like haplotype. To avoid multimapping, we kept only reads that were mapped once at genome, and as a quality filter, we kept only mapped reads with a MAPQ score above 10, resulting in approximately 244 million reads mapped. Visual inspection of bio-logical replicates (S2A Fig) and Spearman correlation analysis (on a genome-wide read cover-age of 100 bp windows) for each replicate (S2B Fig) indicated a high correlation among them, showing good reproducibility and agreement among biological replicates.

Trypanosomatids have an unusual genomic structure due to the gene organization into polycistrons and lack of specific promoter regions for each coding gene. However, the gff file, which describes the genomic features of the *T. cruzi* CL Brener strain available at TriTrypDB, lacks the annotation of some important regions as described below. Thus, we carefully curated this file to extract more meaningful biological conclusions from our dataset, determining both the size and location of polycistrons and their regulatory regions and removing features with sequences with more than 10% gaps (N) from our analysis to avoid inflation of length size for some features. Therefore, we obtained coding DNA sequences (CDSs) (median size, 1118 bp) flanked by short intergenic regions (median size, 590 bp) (S1 Table) forming long polycis-tronic units (PTUs) (median size, 19174 bp) that are transcribed in the same direction. The intergenic region flanking two PTUs plays the role of TSS or TTS if flanked by divergent (dSSR, median size of 1844.0 bp) or convergent (cSSR, median size of 1928.5 bp) PTUs, respec-tively. Genome mapping of histone variants (H2A.Z) and RNA Pol II enrichment as well as transcriptomics analysis of small primary transcripts carrying a 5'triphosphate and nuclear run-on assays indicate the existence of TSSs at a non-dSSRs [26,32,35]. Due to the lack of simi-lar datasets in *T. cruzi*, our strategy was focused only on the TSS found at dSSR.

To obtain the NPO values and compare them between the life forms, we used DANPOS2, which is a robust bioinformatic pipeline, to infer nucleosome mapping and detect dynamic nucleosomes [36]. This latter is inferred by optimizing data normalization and statistical test-ing between two samples. One advantage of DANPOS is that it uses an average read size, cor-recting for differences among MNase digestions and therefore minimizing experimental variation. Fig 1A clearly indicates that *T. cruzi* genomic regions are covered by nucleosomes with clear peaks and depletions in some regions. Approximately 125,000 nucleosomes were identified for each life form (Fig 1B). On average, all features are covered with similar amounts of nucleosomes per Mbp (5427.1 +- 214.7), with slightly fewer nucleosomes in dSSRs and cSSRs than in intragenic and intergenic regions (Fig 1C).

Fig 1B shows that the nucleosome distribution among all genome features is very similar between the life forms. A total of 27,245 nucleosomes (approximately 22% of total) differed between life forms (FDR < 0.05) (S2 Table) and were therefore considered dynamic. When dynamic differences were compared against static nucleosomes of both types (Epi and TCT), significant differences (in terms of chi-square residuals, under $\alpha$ = 0.05) were observed and associated with dynamic positions only for "dSSR" and "other" categories with a larger presence of dynamic nucleosomes (correct z-score = 2.77; with res = +8.5 and 4.1, respectively), while dynamic nucleosomes were underrepresented in the "mRNA" category (res = -3.3) (S3A Fig).

Dynamic nucleosomes can be further classified by DANPOS into three classes according to the type of change (a scheme of dynamic classes is represented in S4B Fig). Those whose occu-pancy level (measurement of local nucleosome density in a cell population) for a given

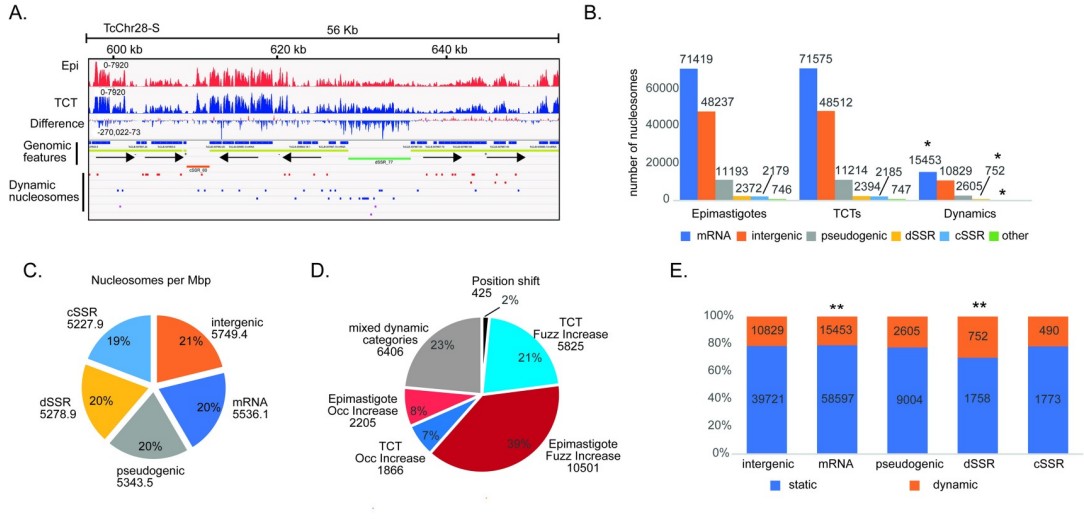

**Fig 1. A.** Representative IGV snapshots of MNase-seq data in *T. cruzi* CL-Brener Esmeraldo-like genome epimastigotes and TCTs. dSSRs and cSSRs are shown in green and orange, respectively. **B.** Nucleosome distribution for each life form and genome feature. The genome is divided into six localization categories: CDS (mRNA), intergenic, pseudogenic, dSSR, cSSR and other, which comprises rRNAs, tRNAs, snoRNAs, snRNAs and chromosome ends. * significant differences using chi-square tests under α = 0.05 with z-score = +/- 2.77 after Bonferroni correction; with residual +8.5 for "dSSR", +4.1 for "other", and -3.3 for "mRNA". **C.** Nucleosome distribution per megabase (Mbp) among all genome features. **D.** Nucleosome distribution of dynamic nucleosomes (FDR< 0.05) according to changes in fuzziness, occupancy level, genomic position shift and mixed dynamic categories. **E.** Percentage of dynamic and static nucleosomes (sum of nucleosomes from both life forms) at the indicated features. ** significant differences using chi-square test under α = 0.05 with corrected z-score = +/- 2.8 by Bonferroni; with residual +9.2 for dSSR and -3.2 for mRNA. In all cases, to obtain reliable lengths for dSSRs, cSSRs and intergenic regions, regions of sequences presenting gaps (N) of more than 10% were removed.

genomic location changes when two conditions are compared are classified as "occupancy (occ) changes"; shifts in nucleosome position (in bp) between two conditions, which does not necessarily indicate an occupancy change, are classified as "position shift"; and finally, changes classified as "fuzziness (fuzz)" reflect the standard deviation of nucleosome position in different cells: a low standard deviation means less fuzziness and more well-positioned nucleosomes, while a high standard deviation means more fuzziness and a dispersed nucleosome pattern [2,36,37]. These classes can be found either individually or together in a nucleosome. Here, more than 1500 dynamic nucleosomes are classified into all classes (S4 and S5 Figs).

Considering only dynamic nucleosomes that exhibit a change in occupancy or fuzziness state between life forms, we observed that epimastigotes have almost two times more dynamic nucleosomes than TCTs (12.706 vs 7.691) (Fig 1D). The majority of nucleosome changes involve changes in fuzziness (60%), followed by changes in nucleosome occupancy (15%) and position shift (2%). Twenty-three percent of changes involve a combination of two or more classes of dynamics. Epimastigotes have almost twice as many dynamic nucleosomes with increases in fuzziness when compared to TCTs; in other words, in TCTs, nucleosomes are more well positioned, which could reflect less RNA Pol II movement along the DNA template. Epimastigotes also have more dynamic nucleosomes with increased occupancy (abundance) than TCTs (2205 vs 1866).

## Nucleosome changes are preferentially detected at dSSRs

To obtain better insights into nucleosome organization, we looked into the details of differences at particular genomic features. dSSR regions contain more dynamic nucleosomes (30%)

(chi-squared test; *p-value* <0.05) than all other genomic features (21%, 21% and 21.6%, respectively, for intragenic, intergenic and cSSRs) (Fig 1E and S6 Fig), supporting the notion that chromatin architecture is important to transcription initiation. This is in line with visual inspection of the differential (epimastigote vs TCT) nucleosome signals at single-nucleotide resolution, which indicated that dSSRs are more enriched in nucleosomes in TCTs (Figs 1A and 2A). Importantly, the majority of dynamic nucleosomes are either increased in occupancy or at fuzziness level in TCTs than in epimastigotes (321 vs 145) (Fig 2B). Thus, although TCTs have an overall smaller dynamic nucleosome increase in both fuzziness and occupancy when compared to epimastigotes (Fig 1D), these changes represent 42% of differences found at dSSRs. In this regard, an increase in occupancy at dSSRs is more evident in TCTs than epimastigotes (161 vs 19 nucleosomes) (Fig 2B), indicating that in TCTs, dSSRs are more occupied by nucleosomes, possibly reflecting less elongation of RNA Pol II through adjacent PTUs. The top 10% dSSR with the highest difference in nucleosome occupancy between life forms (S6B Fig) were compared to the TPM (transcripts per kilobase per million) counts in epimastigotes and TCTs of their flanking polycistrons—obtained from a public transcriptomic study [38]. No clear correlation between higher occupancy in dSSR and lower TPM counts (and vice-versa) was observed (S6C Fig). However, to faithfully evaluate the effect of dSSR nucleosome occupancy on RNA Pol II elongation, a global RNA nascent sequencing analysis such as GRO-seq [39] would be more appropriated as transcriptomic assays reflect the steady-state levels of transcripts. To gain more insights into the dynamics at dSSRs, we evaluated the functional enrichment of GO terms associated with the polycistrons related to these dynamic dSSRs (S3 and S5 Tables). To evaluated that, we analyzed only polycistrons with more than 1 coding gene and with 4 or more dynamic nucleosomes, which comprised 1075 genes in 42 polycistrons (Fig 2C). Interestingly, terms associated with "regulation of nuclear division", "regulation of cell division", "regulation of translation", "chromatin remodeling", "adhesion to a symbiont host cell", and "antigenic variation" were enriched (Fig 2D). Since gene expression is mainly regulated by posttranscriptional mechanisms, these findings suggest an unexpected regulation maintained in *T. cruzi* chromatin reflecting phenotypic differences between replicative and nonreplicative forms, as it will be discussed later.

More striking, 61 out of 69 polycistrons annotated as having only one coding region were enriched with dynamic nucleosomes at dSSRs (S3 Table and S7 Fig). Twenty-nine percent of these polycistrons code for rRNA, tRNA or snRNA genes; however, the remaining are enriched in terms associated with "cell adhesion". In fact, 12 encode TS (trans-sialidase), GP63, mucin or MASP genes. It remains to be investigated whether these represent true polycistrons or misannotated regions, which could be achieved by using TSS markers such as deposition of histone variants, specific histone PTM markers and enrichment of RNA Pol II, as observed in *T. brucei* [26,40]. If this proves correct, nucleosome dynamics could represent an unexpected alternative to regulate specific gene sets.

### Dynamic nucleosomes at intragenic regions

To gain more insights into the nucleosome dynamics inside coding regions, we evaluated the GO terms related to dynamic nucleosomes found at intragenic regions (coded for all classes of RNAs: mRNA, rRNA, tRNA, snoRNA, and snRNA). First, we compared all IDs that have at least one dynamic nucleosome (3499) and those with no dynamic nucleosomes (6980) (S8A Fig and S5 Table). The first category is enriched in GO terms associated with "cytokinesis", "cell cycle", "development process", "tRNA aminoacylation" and "adhesion". To produce a more stringent set of IDs with dynamic nucleosomes, we ranked the IDs by the number of dynamic nucleosomes per kbp and selected the top 100 and the bottom 100 as a control (Figs

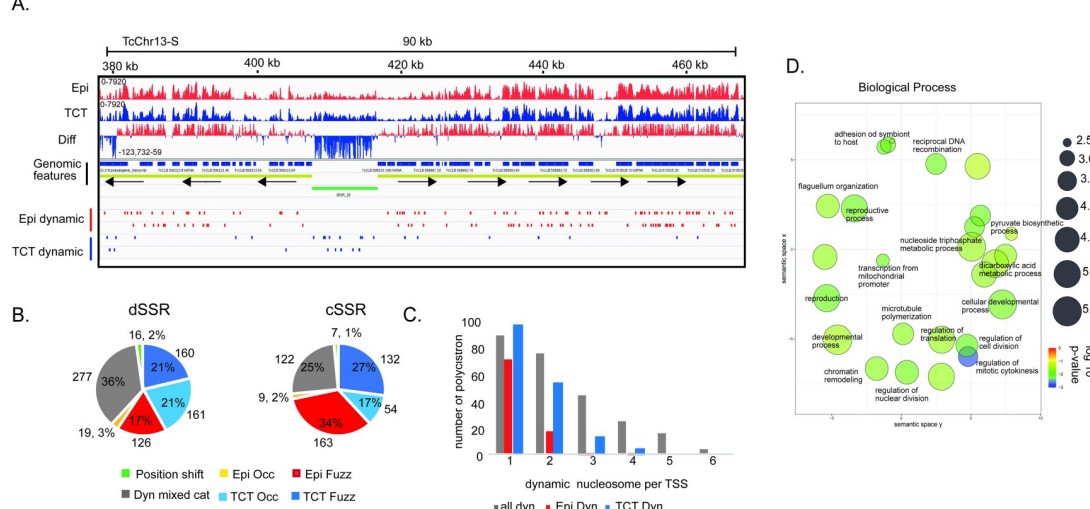

**Fig 2. dSSRs are demarcated by dynamic nucleosomes. A.** IGV snapshots of a representative dSSR (green rectangle) showing an increase in nucleosome occupancy in TCTs. The diff wig file indicates the occupancy difference at each base pair between the epimastigote and TCT data: red means more occupancy on epimastigote and blue means more occupancy on TCT. **B.** Distribution of dynamic nucleosome classes in dSSRs and cSSRs. **C.** Distribution of polycistrons according to the number of dynamic nucleosomes found at their associated TSS. Here, dynamic nucleosomes were obtained from 1000 bp upstream and 500 bp downstream of the first ATG of the CDS of the first polycistron. **D.** A total of 1075 IDs from polycistrons that have at least 4 dynamic nucleosomes at their corresponding dSSR were searched for GO/REVIGO functional annotation using TriTrypDB tools. The scatterplot shows a clusterization of GO terms (remaining after a redundancy reduction) in a two-dimensional semantic space, resulting in similar terms being plotted next to each other according to [75]. In general, more semantically similar GO terms are closer in the plot.

3A and S8B). For this analysis, we excluded IDs classified as hypothetical proteins, as few (or no) GO terms are described for them. Higher fold enrichment in some GO terms (*p-value* < 0.05) was found for the top 100 IDs (Fig 3A). More strikingly, the GO terms "RNA splicing", "regulation of cell cycle process", and "regulation of cell division/cytokinesis" have a fold enrichment of 8 to 40 for the top 100 most dynamic IDs, indicating that many genes/proteins associated with these processes are enriched in dynamic nucleosomes. GO terms associated with the bottom 100 IDs were mostly enriched in terms associated with "ATP hydrolysis" and "energy metabolism/transport", among others. Except for the GO term associated with "translation", no other similarity was found between them (S5 Table).

To further evaluate whether these two categories represent genes with different mRNA transcriptional levels, we evaluated their transcript levels in epimastigotes and TCTs previously obtained by a transcriptomic study [38]. Notably, the top 100 genes, associated with the most dynamic nucleosomes per kbp, had a much wider distribution of transcript level ratios, in accordance with more expression level differences among life forms (F-test <0.001 and Mann-Whitney <0.01) (Fig 3B). Consistent with this observation, 32 of the top 100 IDs were considered differentially expressed (E/TCT fold change > 2 or < -2, p-value <0.05, FDR = 0.01), in contrast to only 18 genes in the bottom 100 genes.

Finally, we wondered whether some GO terms would be preferentially enriched on dynamic nucleosomes in epimastigotes or TCTs. Then, we analyzed the IDs that exhibit increases (in occupancy and/or fuzziness level, FDR < 0.01) in dynamic nucleosomes in a given life form (2016 in epimastigotes and 863 in TCTs) (Figs 3C and S8B). Interestingly, these IDs are associated with different GO terms that are life-form dependent. Terms such as "DNA topological change", "cell division", "mitotic cytokinesis", and "amino acid (or alcohol) metabolic process" were enriched in epimastigotes, while the GO terms "translation" and "TCA

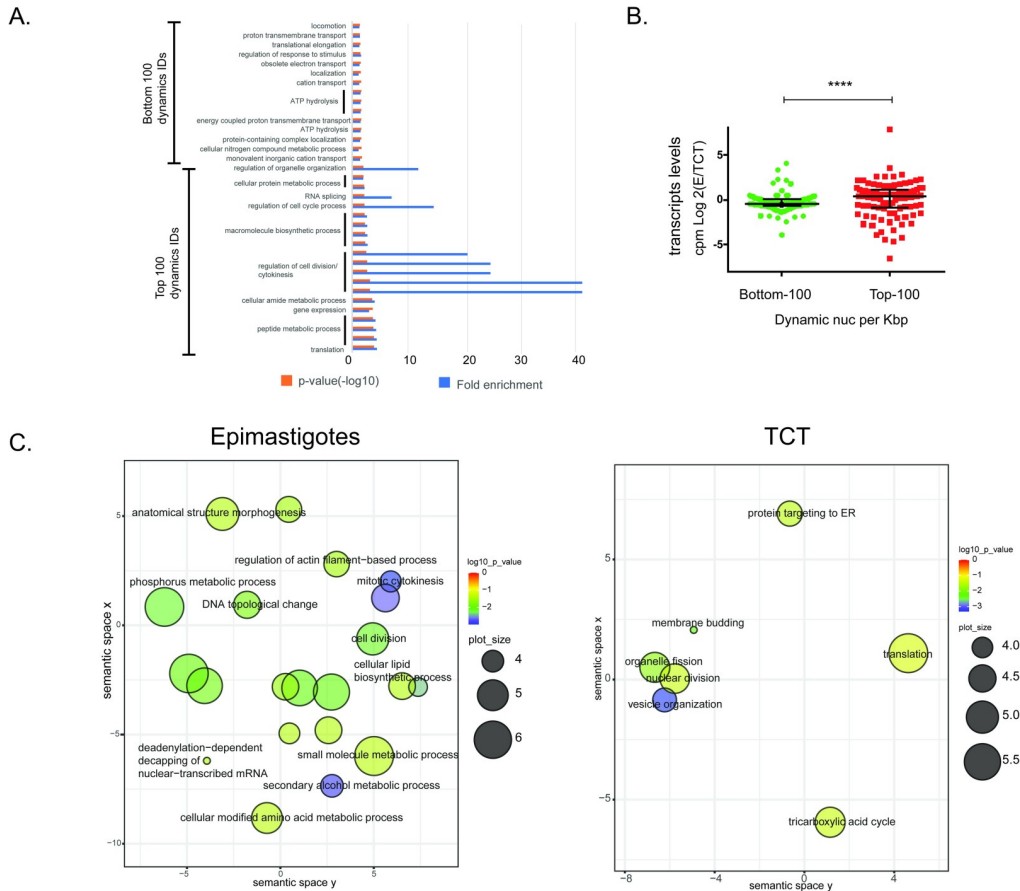

**Fig 3. A.** Biological process GO terms for the 100 most and least dynamic IDs ("top" and "bottom", ranked by number of dynamic nucleosomes per kbp). Fold enrichment and -log$_{10}$ of p-values are shown. **B.** Scatter plot comparing the log$_2$ epimastigotes/TCTs CPM (counts per million) from Li et al [38] transcriptome data for the 100 top and bottom IDs as described in A. Statistical significance was obtained by F-test (p < 0.0001) and Mann-Whitney test (p-value = 0.0018). **C.** IDs that are enriched at fuzziness and/or occupancy specifically, at epimastigote or TCT forms—2016 (left) and 863 (right), respectively—were searched for GO/REVIGO functional annotations using TriTrypDB tools. The scatterplot shows a clusterization of GO terms (remaining after a redundancy reduction) in a two-dimensional semantic space, resulting in similar terms being plotted next to each other according to [75]. In general, more semantically similar GO terms are closer in the plot.

cycle" and "nuclear division" were enriched in TCTs (Fig 3C and S5 Table). Intriguingly, some of these processes are known to be differentially regulated among life forms. For example, TCTs are cell cycle arrested and therefore are not committed to nuclear division or cytokinesis; a global decrease in translation is detected in nonproliferative forms [41]; epimastigotes express more proteins associated with energy metabolism, mainly from the TCA cycle [42]; epimastigotes can release ethanol to the media using glucose as a carbon source [43]; and both amino acid and nitrogen metabolism is enhanced in epimastigotes in comparison to TCTs [42, 44].

Among the IDs with more dynamic nucleosomes in epimastigotes, we highlight two DNA topoisomerases (TcCLB.509203.70 and TcCLB.511589.120), which are enzymes that remove DNA supercoils during transcription and replication: cytokinesis initiation factor 2 (TcCLB.503651.10) and chromosomal passenger complex protein 1 (TcCLB.506221.110), which is the 'master controller' of cell division. Among IDs enriched in dynamic nucleosomes in TCTs, we highlight2 ribosomal proteins (TcCLB.510879.20- ribosomal protein S19 and

TcCLB.511545.40- ribosomal protein L27), 5 40S ribosomal proteins (TcCLB.503833.40, TcCLB.506181.59, TcCLB.506213.60, TcCLB.509825.14, TcCLB.510769.49), 7 60S ribosomal protein (TcCLB.505977.26, TcCLB.506605.150, TcCLB.506861.30, TcCLB.509149.40, TcCLB.509149.60, TcCLB.510309.40, TcCLB.511067.20), 3 translation elongation factors (TcCLB.506599.10, TcCLB.511347.31 and TcCLB.504077.40) and 4 translation initiation factors (TcCLB.506127.170, TcCLB.506693.4, TcCLB.511111.10 and TcCLB.509205.30).

These findings raise the question of whether *T. cruzi* nucleosome organization reflects phenotypic differences observed among the life forms. Nevertheless, it is intriguing, as *T. cruzi* transcription regulation occurs mainly through posttranscriptional mechanisms.

## Nucleosome architecture around specific genomic features

In yeast and some eukaryotes, the nucleosome occupancy around the TSS is very unique, comprising a well-positioned nucleosome upstream of the TSS (-1) followed by a nucleosome depleted region (NDR) and 3 nucleosomes (+1, +2, +3) downstream of the TSS [1]. Higher nucleosome occupancy levels enable the distinction of well-positioned nucleosomes from NDR in each feature landscape. To gain insights into the nucleosome landscape, the MNase-seq coverage per bp was calculated along the -2000 and +2000 bp around the TTS and TSS (as defined in the Materials and Methods section). We found an NDR followed by at least 2 nucleosome-enriched peaks just downstream of the TSS (Fig 4A), along with a fuzziness pattern both upstream and downstream. We noted that the distribution pattern of nucleosome regions around the TSS in epimastigotes and TCTs are similar; however, the nucleosome occupancy abundance, mainly in the upstream region, is higher in TCTs (Fig 4A), which agrees with the increased occupancy of dynamic nucleosomes observed at dSSRs in the latter (Fig 2B). The

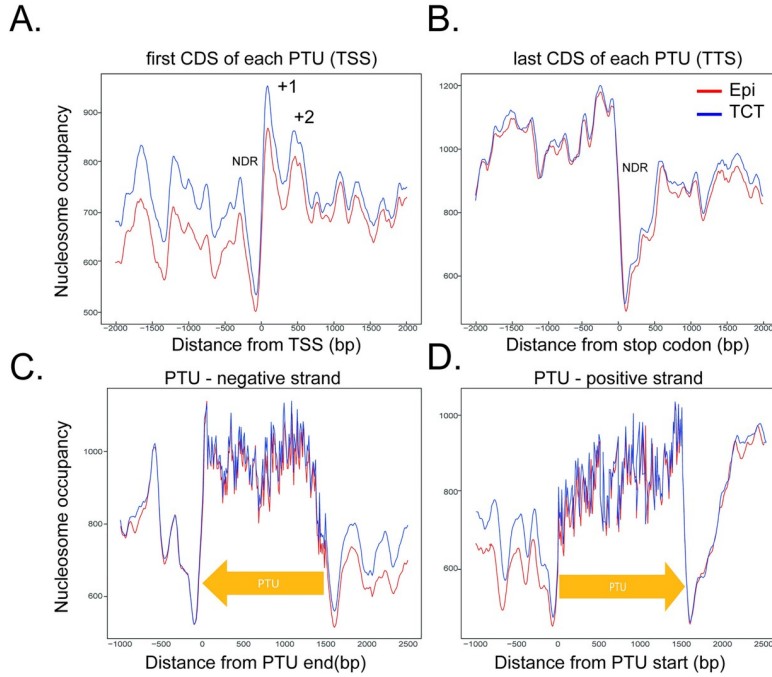

**Fig 4. Nucleosome occupancy along PTUs in epimastigotes (red) and TCTs (blue). A.** Nucleosome occupancy around the first CDS of each PTU. The nucleosome occupancy at the last CDS of each PTU (**B**). **C.** Nucleosome occupancy across negative strand PTUs. **D.** Nucleosome occupancy across positive strand PTUs. In A and D, the first ATG is represented as 0 on the x axis. In B and C, the last base of each CDS is represented by 0 on the x axis.

increase in occupancy at dSSRs can be further observed in Fig 5C. We envisage that this distribution difference reflects differences in global transcription rate observed between the life forms [15]. The nucleosome landscape at the last CDS from each PTU (the putative TTS) is very similar in both life forms, with a clear NDR just downstream of the last stop codon and a decrease in nucleosome abundance in the following nucleotides in comparison with the upstream (and coding) region. Thus, *T. cruzi* Pol II termination sites have an evident NDR, as found in yeast and *T. brucei* [31]. In the latter, a weak NDR was also found upstream of the first gene from each PTU [31,32]. The distinct nucleosome landscape at *T. cruzi* TSSs and TTSs was further evident when they were evaluated in the polycistronic environment (Fig 4C and 4D).

A clear distinction of nucleosomal landscapes for different gene classes (namely, CDS, rRNAs, snoRNAs, snRNA, tRNA) and intergenic regions was observed and was very similar between life forms (Fig 5). Two probable well-positioned nucleosomes were found at the beginnings of rRNA genes (Fig 5D). A single well-positioned nucleosome was found at the snoRNA start codon (Fig 5F). A decay in nucleosome occupancy was observed at the start codons of both snRNAs (Fig 5G) and tRNAs (Fig 5E), suggesting an NDR. An NDR was observed at the first codon (ATG) of each CDS, followed by a well-positioned nucleosome (Fig 5A). In metazoan and plant genomes, nucleosomes with well-defined positions were also found to be associated with splicing sites [45]. In *T. brucei* and Leishmania spp., a well-positioned nucleosome is found at internal splicing sites [30–32], suggesting a role for the nucleosome in defining sites for trans-splicing. Although the 3 and 5' UTRs of *T. cruzi* genes were not mapped, this NDR likely marks the splice site, as observed in other trypanosomes.

Regarding nucleosome occupancy level, there is a clear visual difference at tRNAs and snoRNAs (Figs 5E, 5F, 5H and S9), which exhibit higher and lower occupancy, respectively, in TCTs. Ultimately, this result could clarify the impacts of different RNA polymerase machineries on the nucleosome landscape. CDSs and snoRNAs are transcribed by RNA Pol II, while tRNAs, 5S rRNA and snRNAs are transcribed by RNA Pol III, and the large ribosomal RNAs are transcribed by RNA Pol I [46,47]. Here, we found that the nucleosome landscapes of different gene classes transcribed by different RNA polymerases are indeed different.

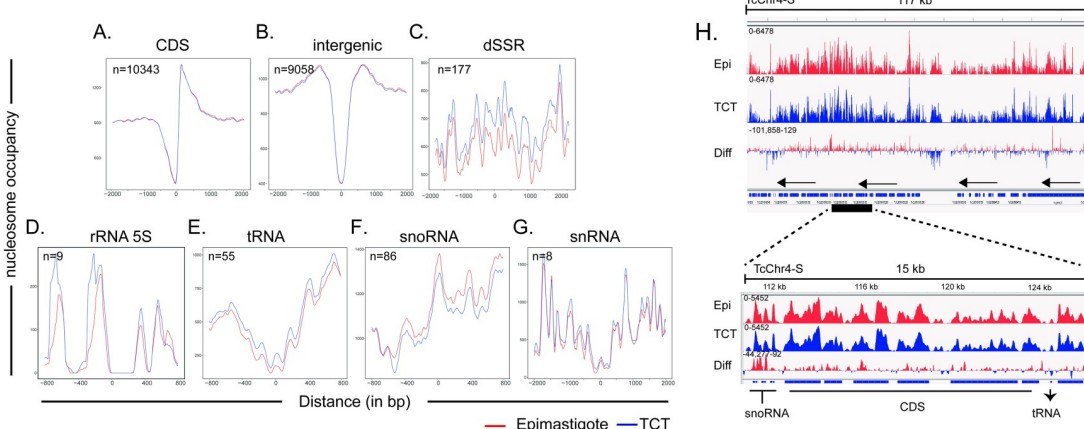

**Fig 5. Nucleosomal occupancy among gene classes and features for epimastigotes (red) and trypomastigotes (blue). A-G.** Occupancy levels were obtained by DANPOS2 using kb ranges (at x axis) for each gene class/feature. The number of elements used for each plot is shown. For dSSRs, landscapes were built from the center region. All other classes, the zero (x axis) represents the first nucleotide for the corresponding gene. Evaluation of occupancy overlap and a Tukey test from distance values obtained from epimastigote and TCT landscapes point by point are shown in S10 Fig. **H.** IGV representation of DANPOS nucleosome peaks highlighting snoRNA, CDS and tRNA regions at TcChr4-S.

To evaluate whether these landscapes differ between life forms, we applied two approaches. First, we calculated the density of nucleosome occupancy (values on the *y* axis from Fig 5A–5G) from all gene classes in epimastigotes and TCTs (S10 Fig). The overlap percentage indicates that dSSRs have the lowest values (0.45), followed by cSSRs (0.7), rRNAs (0.7) and snoRNAs (0.71). The nucleosome landscapes of CDS and intragenic regions are the most similar between epimastigotes and TCTs, with overlaps of 0.8 and 0.89, respectively. To further evaluate differences in nucleosome landscapes/occupancies of different life forms, the distance (in *y* axis) between epimastigote and TCT landscapes were measured and plotted in S10B Fig. A Tukey test was performed (considering intergenic regions as a control with no difference between life forms) to statistically identify gene classes with different nucleosome landscapes/occupancies between life forms. Again, dSSR was considered the most different, followed by tRNA, snRNA, snoRNA, cSSR and rRNA (S10C Fig).

## Nucleosome positioning on multigenic family members

*T. cruzi* has a very repetitive genome encoding many multigenic families, including transialidase (TS), GP63, mucin, MASP, DGF-1 and RHS families [11]. In general, these genes are very polymorphic, encode cell surface antigens and virulence factors (except RHS) and are mainly expressed in infective forms [42,48]. Recently, the *T. cruzi* genome was proposed to be divided into a core compartment and a disruptive compartment with differences in GC and gene content mainly related to the distribution of the multigenic families [49]. Due to the biological importance and genomic peculiarities of these groups of genes, we investigated their nucleosome organization in detail.

From the selected dynamic nucleosomes with FDRs lower than 0.01 (totaling 3499 IDs, or 33% of the total IDs), we verified that 80% of DGF-1 genes have at least one dynamic nucleosome (Fig 6A), which is in clear contrast with other gene classes. Interestingly, this is not due to a larger total length (in bp) of DGF-1 genes, as TS genes occupy as much of the genome as DGF-1 genes (S11A Fig). In addition, when normalized by total length in kbp, DGF-1 genes had more dynamic nucleosomes per kbp than all other gene classes (S11A Fig). Recently, we detected that replicative origins are preferentially located at DGF-1 genes, and we proposed that genetic variability may be induced due to frequent collisions between replication and transcription machineries [50]. Here, we detected that this gene class is also more likely to have dynamic nucleosomes (S11B Fig).

We counted the number of dynamic nucleosomes per multigenic family member and compared them with disrupted and conserved compartments (S11C and S11D Fig). While 20% of nucleosomes that cover intragenic regions are located at multigenic family members, the same number for dynamic nucleosomes is 31% (Fig 6B), an increment of 1.7 times (chi-squared test; *p-value* <0.05). TS, DGF-1 and RHS are overrepresented at dynamic nucleosomes when compared to all nucleosomes distribution at the genome (chi-squared test; *p-value* <0.05– S3C and S3D Fig). DGF-1 and TS had the most significant differences, with increases of 1.6 and 2.4 times, respectively (Fig 6B). Taken together, these data suggest that these members of the multigenic family are more prone to changes in NPOs.

As highlighted previously (Fig 1D), TCTs have fewer dynamic nucleosomes than epimastigotes (12.706 vs 7.691); however, the distribution of these dynamic nucleosomes in intragenic regions is not even between life forms (chi-squared test; *p-value* <0.05) (Fig 6B). Dynamic nucleosomes at multigenic family members represent 29.7% of changes in epimastigote intragenic regions but 36.2% of changes in TCTs. Statistical analysis (chi-squared test; *p-value* <0.05) indicated that members of the DGF-1 family were enriched in dynamic nucleosomes in TCTs (S3D Fig).

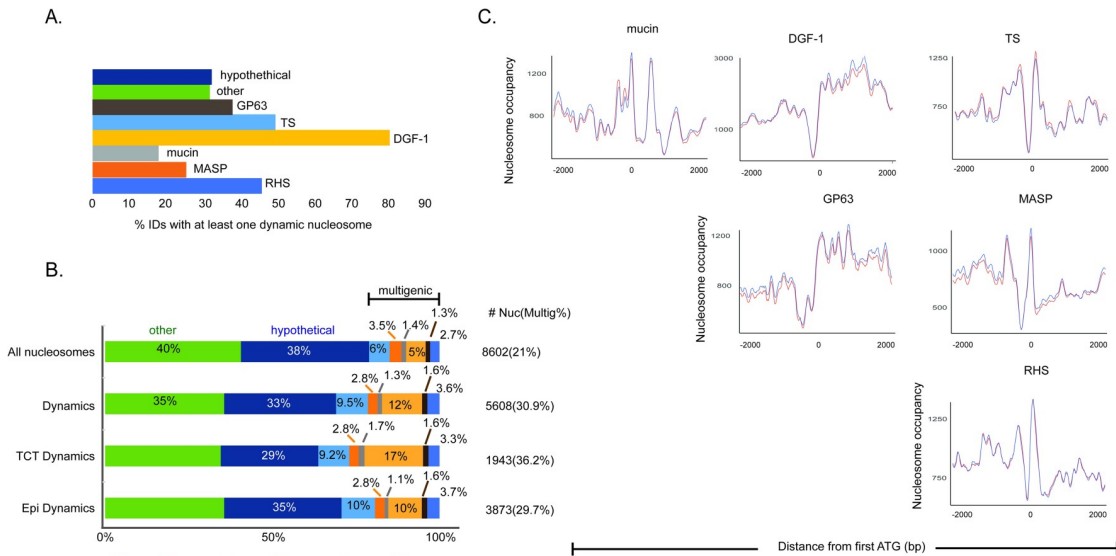

**Fig 6. Nucleosome organization of multigenic family members. A.** Percentage of the indicated genes with at least one dynamic nucleosome (FDR <0.001). **B.** Distribution of all nucleosomes, dynamic nucleosomes, and dynamic (increase at occupancy and/or fuzziness) nucleosomes in epimastigotes and TCTs at intragenic regions (all genes classified as "mRNA", "pseudogenic", "tRNA", "snoRNA", "snRNA", "rRNA"). The "other" category refers to all intragenic regions except those represented here. A significant difference was observed for categories "All nucleosomes" vs "Dynamics" (p-value <0.05, chi-square test under α = 0.05 with corrected z-score = +/- 2.7 by Bonferroni), with significant positive residuals (overrepresentation on dynamic nucleosomes) for categories TS (+10.2), DGF-1 (+19.1) and RHS (+3.7), while negative residuals (underrepresentation on dynamic nucleosomes) were observed for hypothetical (-5.9) and "other" (-5.9). A significant difference was observed for categories "epimastigotes" vs "TCT" (p-value <0.05, chi-square test under α = 0.05 with corrected z-score = +/- 2.7 by Bonferroni), with significant positive residuals (overrepresentation on TCTs) for categories DGF-1 (+6.2) and negative residuals (underrepresentation on epimastigotes) for DGF-1 (-4.3). Full chi-square test residuals values are given in S3 Fig. **C.** Nucleosomal landscape among genes from the multigenic family for epimastigotes (red) and trypomastigotes (blue). Occupancy levels were obtained by DANPOS2 using kb ranges (at x axis) for each gene class.

The nucleosome landscape of members of the multigenic families differs from that of the other CDSs transcribed by RNA Pol II (Fig 5B), as shown in Fig 6C. MASP, TS and RHS genes have a similar landscape, with a clear NDR near the first ATG followed by a well-positioned nucleosome downstream and a fuzziness pattern. On the other hand, DGF-1 and GP63 are more similar to each other, with an NDR just upstream of the first ATG, followed by an increase in nucleosome occupancy all along the gene body. Curiously, DGF-1 and GP63 can be found in both disrupted and conserved compartments [49]. Mucins show a nucleosome landscape that looks different from both groups. Taken together, these data suggest a nonrandom landscape of nucleosomes in *T. cruzi* and a peculiar NPO at multigenic family members.

## Evaluation of nucleosome occupancy and expression levels

We asked whether the nucleosome landscape would reflect Pol II transcription activity. Thus, we classified transcripts into 4 groups (high, medium, low and differential) based on their expression levels in epimastigotes and TCTs obtained in a public transcriptomic study [38]. Although this strategy analyzes only the steady-state levels of transcripts, which are the combined result of transcription, RNA processing, and mRNA degradation, we observe that the nucleosome occupancy levels differ among genes in accordance with their expression levels (Fig 7). In brief, the more highly expressed a gene is, the more pronounced the NDR and the more well-positioned the nucleosome downstream of it. Additionally, the association of the nucleosome landscape with expression levels occurs both in epimastigotes and TCTs.

Finally, we asked whether the differentially expressed genes had different nucleosome landscapes in different life forms. Interestingly, the pattern is very similar to that of the low expressed genes in both TCTs and epimastigotes (Fig 7).

## Discussion

Trypanosomes are valuable models to understand the evolution of chromatin architecture because the Excavata subgroup, in which trypanosomes belong, is considered one of the

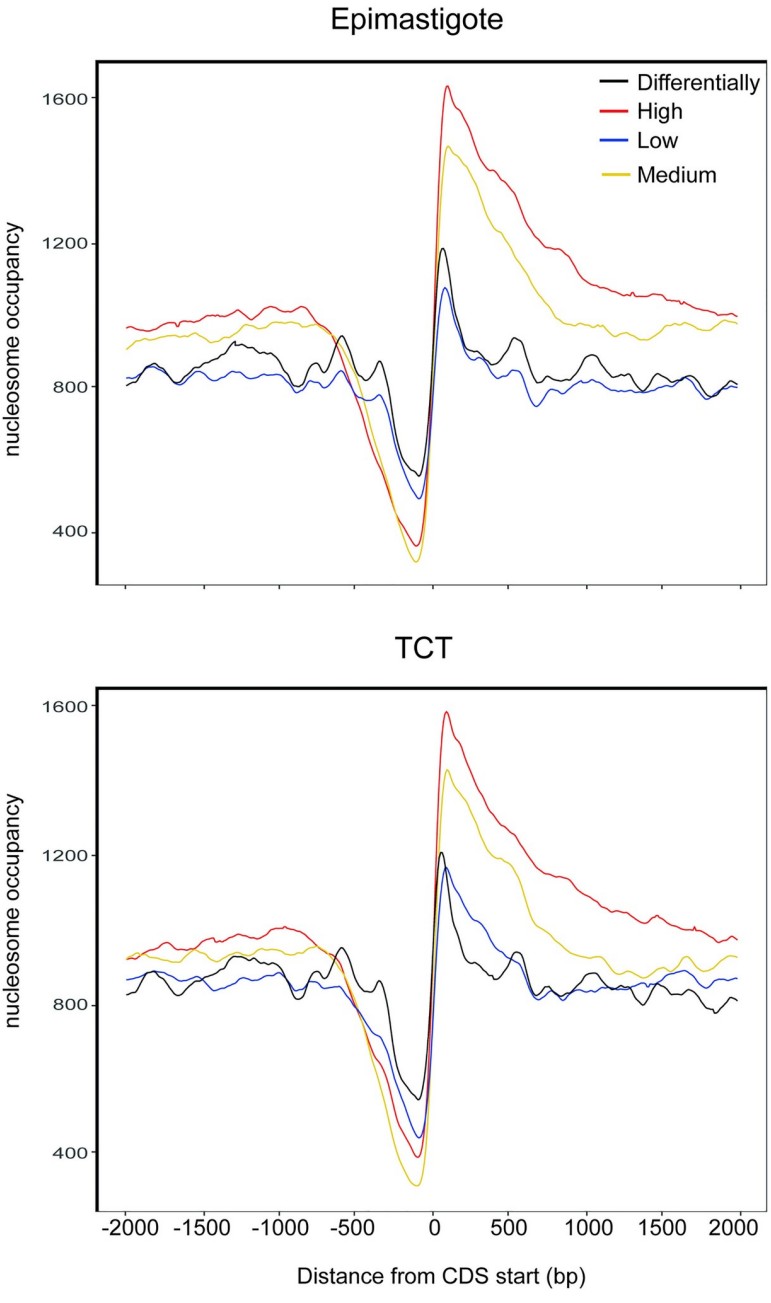

**Fig 7. Nucleosomal occupancy according to gene expression level.** High (red), medium (yellow), low (blue) and differentially expressed (black) genes for epimastigotes (top) and TCTs (below). The first ATG from each CDS is shown as 0 on the x axis. The number of CDSs used in this analysis was as follows: for epimastigotes (high, 3485; medium, 3484; low, 3353; differential, 1872) and for TCTs (high, 3486; medium, 3484; low, 3352; differential, 1663).

deepest branch of eukaryotes [51]. In addition, the lack of a canonical regulation of gene expression associated with a marked reduction in transcription factors [52] makes trypanosomes interesting tools for investigating the impact of epigenetic regulation on transcription regulation. In addition, epigenetic alterations are targeted by environmental changes, and trypanosomes, as digenetic organisms, must adapt to different host environments to complete their life cycle. As highlighted previously, epimastigotes and TCTs have profound differences in cell morphology, infectivity, DNA replication, and global transcription rates [15,42,53]. Most importantly, their differences in DNA replication and transcription may shed further light on the impact of these events in shaping the nucleosome landscape. Previous analysis on replication and transcription capacity in trypanosomes focused on either analyzing in one life form [30,32] or in two life forms with similar phenotypes regarding replication and transcription capacity [31].

Here, we used a powerful pipeline that allowed us to faithfully compare the nucleosome positioning and occupancy across *T. cruzi* life forms by detecting dynamic nucleosomes at strategic genomic regions. The most remarkable findings were as follows: i. dSSRs are more affected by nucleosome dynamism than all other genomic features and are more enriched in nucleosomes mainly in TCTs; ii. multifamily genes, mainly associated with infective stage and virulence factors, have more dynamic nucleosomes; we highlight TS, RHS and especially the DGF-1 family genes, 80% of which contain at least one dynamic nucleosome; iii. dynamic intragenic regions and PTUs transcribed by dynamic dSSRs encode genes related to critical regulatory processes, such as cytokinesis, translation, gene regulation, RNA splicing and cell adhesion; and iv. the nucleosome landscape and dynamics reflect gene expression. As highlighted previously, our computational strategy considered only TSSs located within dSSRs. Thus, it remains to be elucidated whether TSSs located at non-dSSRs also harbor more dynamic nucleosomes.

The presence of nucleosomes in a given DNA sequence affects its accessibility; therefore, the composition and organization of nucleosomes at TSSs and replication origins are tightly regulated and play a fundamental role in controlling transcription and DNA replication [54,55]. These processes, as well as DNA repair, disrupt local nucleosome organization that is restored by different proteins, including chaperones. For example, transcriptional elongation by RNA Polymerase II may shape chromatin structure by promoting the displacement of nucleosomes during its passage [56], changing the NPO [4]. Considering the polycistronic transcription scenario in trypanosomes, together with critical changes in global transcription levels among life forms [15], the fact that almost 2 times more dynamic nucleosomes were found in the epimastigote form than TCTs seems to mirror these processes. Importantly, the great majority of dynamic nucleosomes exhibit changes in fuzziness status, which measures the standard deviation of nucleosome positioning in a population [36]. We believe that the more frequent elongation of RNA pol II along PTUs in epimastigotes would generate nucleosomes with slightly different positions, explaining why this life form has more dynamic nucleosomes than TCTs. In addition, the increased dynamics of epimastigotes may also be attributable to the fact that these cells are in different cycle phases, as some nucleosome organization may be slightly different along the cell cycle [57]. Here, we used epimastigote cells in the exponential growth phase, which contains parasites in different cell cycle phases, whereas TCTs are uniformly cell cycle arrested.

We can clearly observe an NDR just upstream and downstream of the first ATG and the last stop codon, respectively, likely associated with splicing sites, as seen in Leishmania and *T. brucei* [30–32]. Moreover, the nucleosome landscape is related to the steady-state level of transcription: the higher the expression is, the higher the NDR depth and the more strongly positioned the nucleosome. Trypanosomatids have no (or very few) introns [58, 59]; however,

trans-splicing is a very frequent event as all transcripts receive at their 5' a spliced leader mini-exon from a separated transcript (reviewed in [10]). The efficiency of trans-splicing can regulate the final transcript levels such that highly expressed genes may have a higher NDR (and therefore a strongly positioned nucleosome), correlating with a more efficient trans-splicing event, as previously proposed by [32].

Promotor regions are typically more depleted in nucleosomes in eukaryotes (reviewed in [5]). Due to the polycistronic gene organization in trypanosomatids, no defined promoters are associated with an individual gene; however, dispersed promoters rich in GT elements were found at dSSRs and were shown to both drive transcription initiation and histone H2A.Z deposition [32]. Here, we found that the nucleosome architecture at dSSRs greatly differs between epimastigotes and TCTs, with the latter enriched in nucleosomes. Previously, it was shown that profound changes in global transcription levels occur between these life forms. In addition to a global nuclear and chromatin rearrangement in parasite life forms with disruption of important chromatin domains [15,53], the underlying molecular mechanism of how exactly this is achieved has been poorly explored. Our findings contribute a piece of this puzzle, showing that dSSRs have higher nucleosome occupancy in TCTs than in epimastigotes, which is likely associated with the lower global levels of transcription in TCTs. It remains to be investigated whether the increase in nucleosome occupancy hampers the assembly of RNA Pol II machinery or whether lower levels of RNA Pol II bound to DNA promote higher nucleosome occupancy. In addition, our data indicate that different dSSRs may have a different chromatin structure that may generate PTUs with different transcription levels.

Although the genes located in polycistrons have unrelated functions, it was surprising that dSSRs that contain more dynamic nucleosomes are associated with polycistrons whose genes are enriched in terms associated with known phenotypic differences observed between life forms (Fig 2). It is well known that the differentiation of replicative forms to nonreplicative forms is associated with a global decrease in translation activity, an increase in the capacity of adhesion to substrate/cellular epithelia, changes in nuclear and chromatin architecture, and changes in the expression of membrane-associated proteins, in addition to ceasing cellular division [15,41,42,60, 61]. Interestingly, many terms and genes associated with these events were detected in the GO analysis, which is depicted in Figs 2D, 3 and S8 (S4 and S6 Tables). Among the genes, we found the DOT1B (TcCLB.511417.70) enzyme, which is responsible for the trimethylation of H3K76. We have previously shown that multiple methylation states at H3K76 are cell cycle dependent [25]. TCTs have predominantly H3K76 in a trimethylated form, while replicative forms (epimastigotes) have also mono- and dimethylated H3K76, which are associated with G2 and G2/mitosis, respectively. In addition, regulator of chromosome condensation (RCC1- TcCLB.506885.416), condensins, and proteins with crucial roles in preparing chromatin for mitosis were also enriched in dynamic nucleosomes, a cell division protein kinase (TcCLB.504125.90) and centrins (TcCLB.508323.70 and TcCLB.508323.60) (S4 and S6 Tables). Along these lines, when analyzing individual genes with more dynamic nucleosomes (Figs 3 and S8), a unique functional enrichment was observed. Genes such as cytokinesis initiation factor 2 (TcCLB.503651.10) and the chromosomal passenger complex protein 1, both associated to mitotic cytokinesis were found enriched in dynamic nucleosomes in epimastigotes. In addition, many terms related to metabolic process appeared enriched. Transcriptomic analysis of *T. cruzi* life forms indicates a global metabolic switch among *T. cruzi* life forms [42].

Recently, the *T. cruzi* genome was proposed to be organized in disrupted and conserved compartments based on the distribution of multigenic family members [49]. Here, we observed that the RHS, TS and DGF-1 genes have more dynamic nucleosomes than other classes. Most striking is the fact that at least one dynamic nucleosome is found in 80% of DGF-1

genes. Recently, we detected that replicative origins are preferentially located at these genes, and we suggest that this positioning may induce genetic variability due to frequent collisions between replication and transcription machineries [50]. It remains to be investigated whether DGF-1-containing replicative origins have more dynamic nucleosomes than those that do not harbor origins. Nevertheless, the fact that the majority of these genes have dynamic nucleosomes is interesting and opens new avenues related to the regulation of replicative origins by nucleosome occlusion. Another intriguing fact regards the nucleosome landscapes for multigenic family members that greatly differ from other CDSs transcribed by RNA Pol II (compare Fig 5A and Fig 6). Even within multigenic family members, GP63 and DGF-1, which transit between both conserved and disrupted genome compartments, have similar nucleosome landscapes, while TS, mucin and RHS are similar to each other. DGF-1 has the highest GC content compared to all other genomic features, including other members of the multigenic family. Whether this composition may play a role in the increased nucleosome dynamism or on the peculiar nucleosome landscape needs to be further evaluated; however, *in vitro* studies indicate that the high GC content tends to associate with compact nucleosomes with more affinity to DNA [2]. The association of chromosome location and nucleosome landscape also needs to be further evaluated, as TS and DGF-1 are mainly located in subtelomeric regions, which are believed to be enriched in heterochromatin [62].

The DNA base composition is important for nucleosome formation because homopolymers are intrinsically rigid and therefore hamper nucleosome formation [63]. Our findings open the question of how the observed difference in NPOs is achieved between life forms. As epimastigotes and TCTs have the same cis elements, one hypothesis is that the more dynamic regions may be related to an uneven deposition of histone variants between life forms, forming more unstable nucleosomes in some features [26], such as at dSSRs and in members of the multigenic families. In fact, we previously observed that H2BV (the partner of H2A.Z) is enriched in TCT chromatin [25]. We are currently investigating the genomic location of H2BV; however, it is tempting to propose that enriched dynamic regions may have a different content of H2BV in different life forms. We have previously shown that TCTs have lower global levels of PTMs than epimastigotes and metacyclics with some PTMs, such as those from the H4 N-terminus, greatly different among them [23,25]. In addition, histone H1, which associates with linker DNA, is more phosphorylated in TCTs than epimastigotes and is differentially regulated throughout the cell cycle [64,65]. As phosphorylation and acetylation weaken histone-DNA association, the differential histone PTMs at some genomic features (as well as along life cycle) may influence local chromatin structure. Lastly, nucleosome organization may be affected by different histone turnovers, considering that histone synthesis is regulated by the cell cycle [66] and TCTs are cell cycle arrested.

Taken together, the enriched dynamism at these dSSR and intragenic positions would indicate that the nucleosome organization reflects phenotypic differences at *T. cruzi* even in a context where gene expression is regulated mainly by posttranscriptional mechanisms [10]. One possibility (as discussed above) is that the nucleosome organization of these regions may guide different efficiencies of trans-splicing (occurring in close contact with chromatin), which may be important for modulating the final transcript levels. Another remote possibility is specific gene regulation at these regions. As shown by Kolev et al [58], there are some transcription start sites inside polycistrons, suggesting that RNA polymerase can initiate transcription directly on some gene sets. Nevertheless, it is interesting to find that intragenic regions associated with more dynamic nucleosomes indeed have more differences in transcription levels among life forms.

Our results indicate that *T. cruzi* NPO has critical differences at regulatory regions associated with transcription initiation and splice acceptor sites, at multigenic family members

(mainly at DGF-1) and at some intragenic regions reflecting phenotypic differences observed along life form. As trypanosome gene expression relies on posttranscriptional mechanisms [10], how it is achieved and whether it indeed has a functional significance remain to be investigated. Finally, our results indicated that phenotypic expression may reflect nucleosome organization even in a context with a lack of canonical transcriptional control.

## Materials and methods

### Cell culture

*T. cruzi* (CL Brener strain) epimastigotes were cultured in liver infusion tryptose (LIT) at 28°C at a density of $3x10^6$ parasites/ml. To obtain trypomastigotes from the supernatant of host cells, LLC-MK2 cells were cultured in Dulbecco's modified Eagle's medium (DMEM) supplemented with 10% fetal bovine serum (FBS) at 37°C and 5% $CO_2$ during all infection assays. The metacyclic trypomastigotes (previously obtained by *in vitro* differentiation from epimastigote forms, following [67]) were added to the cell culture and left for 24 hours. The medium was changed every day until the end of the experiment. At two weeks after infection, the supernatant was collected and centrifuged at 2500 rpm for 5 minutes and kept at 37°C for at least one hour. To collect just the trypomastigotes present in the supernatant, the medium was removed carefully, the parasites were centrifuged at 4000 rpm for 10 minutes, and the pellet was frozen at -80°C until the day of the experiment.

### Nucleosomal DNA extraction

*T. cruzi* epimastigotes and trypomastigotes ($10^8$ parasites) were centrifuged at 4000 rpm for 5 minutes, and then the pellet was washed in lysis buffer (1 mM potassium L-glutamate, 250 mM sucrose, 2.5 mM $CaCl_2$, 1 mM PMSF). After centrifugation, parasites were lysed in lysis buffer with 0.1% Triton X-100 and centrifuged again, and the supernatant was discarded. Additional washes with lysis buffer without detergent were performed, and the samples were resuspended in 150 μl of lysis buffer, incubated with *micrococcal nuclease* (Thermo–Part Number 88216) (MNase, 1500 U) for 30 minutes at 37°C and then supplemented with 200 μg proteinase K (20 mg/ml) and incubated for an additional 3 hours at 56°C. The nucleosomal DNA was extracted from each group by the phenol-chloroform method, purified from a 1.5% agarose gel using the GFX PCR DNA and Gel Band Purification kit (GE Healthcare) and analyzed in a Bioanalyzer 2100 instrument (High Sensitivity DNA assay, Agilent Technology). The procedures were conducted in biological triplicates for epimastigote and trypomastigote samples, which were sent for library preparation and Illumina sequencing, at the University of Glasgow (Polyomics Glasgow, Scotland) using the New England Biolabs NEBNext Ultra II DNA library prep kit for Illumina.

### Sequence read processing, alignment and NPO identification

The methods applied to analyze the MNase-seq data are depicted in S1B Fig and described below. DNA fragments were deep sequenced using Illumina NextSeq 500. Paired-end reads (2 x 75 bp) were quality checked with FASTQC and filtered by Trimmomatic [68] using the following parameters: windowSize 15, requiredQuality 25, minlen 35; sequencing adapters were also removed from reads. Filtered reads were then mapped against the CL-Brener Esmeraldo-like genome (downloaded from https://tritrypdb.org/-DB32) using Bowtie 2.2.9 [69] individually for each replicate. The alignment files were filtered using a MAPQ (MAPing Quality) score of 10 using samtools [70] to avoid random read mapping. Spearman correlation analysis was performed by applying multiBamSummary and plotCorrelation from deepTools 3.3.0 [71]

using 100 bp windows to estimate genome-wide read coverage among the mapped sequencing samples.

Nucleosome positioning and occupancy were detected by DANPOS2 [36] using the function Dpos (—paired 1,—clonalcut 1e-10) on pooled triplicates of epimastigotes and trypomastigotes. To ensure a proper comparison, the data were quantile normalized by applying the Wiq function from DANPOS2 and then passed again to Dpos to produce the final MNase peaks. In short, DANPOS2 calculates nucleosome occupancy in two samples and performs a differential test (Poisson by default) to detect signal variations at a given genomic location. MNase peaks were mapped to genomic features (intergenic, mRNA, pseudogenic, rRNA, snoRNA, sRNA, tRNA, TSS and TTS) by BedTools 2.29.0 [72]. Additional custom shell scripts were used to distinguish between the three classes of dynamic nucleosomes presented by DANPOS2 (fuzziness change, occupancy change and position shift). To obtain nucleosomes containing only one dynamic class, the corresponding columns smt_diff_FDR (occupancy change), point_diff_FDR (position shift), and fuzziness_diff_FDR (fuzziness change) from the output files were filtered using FDR $\leq$ 0.05 or FDR $\leq$ 0.01. In this way, only the dynamic nucleosome class column needed to have the desired FDR value, while ensuring the others would present a higher FDR value (e.g., to obtain only nucleosomes with occupancy change class, the smt_diff_FDR needed to be $\leq$ 0.05, while point_diff_FDR and fuzziness_diff_FDR values were > 0.05). The additional column treat2control_dis was used to obtain the position shift class, selecting values > 0 and applying the same logic described before. For dynamic nucleosomes carrying multiple classes, a mixture of these parameters was used. Only dynamic nucleosomes carrying one class of occupancy or fuzziness change were considered for a given life form. The diff wig output was used to extract nucleosome occupancy difference between Epimastigote and TCT. The dSSR bed file (depicting their genomic coordinates) was used, along with the pyBigWig library, to extract the occupancy difference values for each dSSR and to calculate their average value.

Nucleosome occupancy profiles were generated by the profile function of DANPOS2 using the bed files from MNase peaks, features obtained in the genome annotation gff file and transcriptome information. The results from DANPOS2 were visualized in IGV (Integrative Genomics Viewer) [73].

The datasets generated and/or analyzed during the current study are available in the NCBI BioProject repository, under the accession code PRJNA665060 and BioSample codes SAMN16241276, SAMN16241277, SAMN16241279, SAMN16241280, SAMN16241281 and SAMN16241282

## Transcriptomic analysis

Transcriptome data were retrieved from the supplementary material of LI et al. [38]. The log-transformed quantile-normalized counts per million expression values for *T. cruzi* genes present in S2 Table (columns HPGL0249, HPGL0250 and HPGL0251 for trypomastigotes and HPGL0252, HPGL0253 and HPGL0254 for epimastigotes) were used to determine the high, medium and low expressed genes. Values were ordered from highest to lowest and posteriorly divided into three groups containing the same number of elements. The first group, containing the highest values, was defined as the highly (high) expressed genes, while the second group was labeled as moderately (medium) expressed genes, and the third group was categorized as weakly (low) expressed genes. Differentially expressed genes were retrieved from S5 Table. A bed file containing all polycistron genomic coordinates and the normalized TPM wig files of the transcriptomic study [38] were used to obtain polycistronic TPM counts by using the annotatePeaks function from HOMER [74].

### Definition of genome features and GFF curation

dSSRs (which comprise TSSs) were defined as being between the first ATGs in the first genes of the two divergent polycistrons. Similarly, cSSRs (which comprise TTSs) were defined as being in the region between the last stop codons of the last genes of two convergent polycistrons.

To obtain dynamic TSSs, nucleosomes obtained from 1000 bp upstream and 500 bp downstream from the first ATG of the first CDS were evaluated. For calculation of genome feature size, regions of sequences presenting gaps (N) of more than 10% were removed. The percentage of gaps was obtained by counting the number of Ns for each individual genome feature using the genome fasta file and gff file and then dividing them by the corresponding individual feature length in bp.

### Functional gene analysis

Gene enrichment analysis was performed using functional annotation tools available at Tri-TrypDB using the default options.

**Chi-square tests.**  In order to test for differences in the distribution of nucleosome characteristics among different life forms, chi-square tests of homogeneity were applied. Results are given in terms of both general table p-value and individual treatment p-values. Residuals (i.e., the life form effect size) are also reported. Comparisons with significant *p-value* (obtained after familywise Bonferroni p-value correction, to be more conservative) and their associated residuals are shown in modified correlation plots generated in R (R Core Team, 2017).

**Pairwise analyses for nucleosome landscape.**  Box-plots for the distribution of occupancy across the intended window size are given. Density plots, for each pair of treatments being compared, were obtained in R. The area overlap between the two density plots for each case was estimated using the library *overlapping*. Pairwise corrected Tukey tests were performed to check for similarities and differences of the nucleosome positions between treatments.

## Supporting information

**S1 Fig.**  A. Electropherogram of MNase digestion of 3 biological replicates from epimastigote and trypomastigote life forms. B. Scheme of the pipeline used to explore the MNase-seq data. C. Number of mapped reads against the *Trypanosoma cruzi* CL-Brenner Esmeraldo-like genome for each dataset (biological replicates). Blue bars represent total dataset reads; orange bars indicate the mapped reads, and gray bars show the remaining mapped reads with MAPQ scores above 10. Percentage is relative to the total reads in each dataset (blue bars).
(PDF)

**S2 Fig.**  A. Representative IGV snapshots of MNase-seq data mapped against the *T. cruzi* CL-Brener Esmeraldo-like genome (TcChr13S-386,961–415,664). Peaks represent the nucleosome occupancy level calculated by DANPOS2. Epimastigote biological replicates (R1, R2 and R3) are shown in red, and trypomastigote biological replicates (R1, R2 and R3) are shown in blue. Merged datasets of each life form are in the first two lines. The last line indicates the difference in occupancy: red peaks indicate high occupancy in epimastigotes, while blue peaks indicate higher occupancy in trypomastigotes. B. Spearman correlation of read counts for each dataset (biological replicates) from epimastigote (Epi) and trypomastigote (Trypo) life forms generated with DeepTools.
(PDF)

**S3 Fig. Scheme of residual values from the chi-square test with Bonferroni correction (p-value <0.05) for data from Figs 1B (A), 1E and S6 (A), S4B (C) and S6B (D and E).** Only

statistical residual values are represented.
(PDF)

**S4 Fig. Distribution of dynamic nucleosomes according to their dynamic changes.** A. Venn diagram of the three dynamic nucleosome classes found by DANPOS2. B. Scheme of dynamic nucleosome classes obtained by DANPOS2. C. Number of dynamic nucleosomes (FDR <0.05) distributed in four genomic features for each life form according to nucleosome dynamic class. On the right, the total number in each category is shown. Here, pseudogenes were not included in intragenic regions. C. Distribution of dynamic nucleosomes classified into 3 categories in intragenic and intergenic regions. A similar distribution for dSSRs and cSSRs is shown in Fig 2B.
(PDF)

**S5 Fig. Occupancy plots for each dynamic nucleosome class are presented in S4B Fig.** Changes in fuzziness (A), occupancy levels (B) and position shift (C) are depicted. Distances are based from -200 bp to +200 bp of the dyad (represented by 0 on the x axis). Trypomastigotes are represented in blue, and epimastigotes are represented in red. D. Histogram showing the frequency of position shifts detected between life forms.
(PDF)

**S6 Fig. A.** Percentage of dynamic and static nucleosomes (sum of nucleosomes from both life forms) at the indicated features. * significant differences using chi-square test under $\alpha = 0.05$ with corrected z-score = +/- 2.8 by Bonferroni; with residual +3.7 for snoRNA (overrepresented at dynamic category). **B.** The diff wig file obtained from DANPOS2 analysis was used to extract nucleosome occupancy difference between Epimastigote and TCT. The dSSR bed file produced in this work was used, along with the pyBigWig library, to extract the occupancy difference values for each dSSR region and to calculate their average value, which are plotted in this histogram. **C.** TPM counts from epimastigotes and TCTs polycistrons were obtained from Li et al (2016). The top 10% dSSR with highest difference in nucleosome occupancy (obtained from B) is highlight in red (epimastigote>TCT) and blue (TCT>epimastigote).
(PDF)

**S7 Fig. Distribution of polycistrons coding for only one gene according to the number of dynamic nucleosomes found at their associated TSS.** Here, dynamic nucleosomes were obtained from 1000 bp upstream and 500 bp downstream from the first ATG of the CDS of the first polycistron.
(PDF)

**S8 Fig. A.** GO terms (biological process) enriched (p<0.05) for IDs with static (FDR< 0.01) nucleosomes (6980) and for IDs with dynamic nucleosomes (3499). For static IDs, all GO terms are shown, while for dynamic IDs, only the top 20 terms are shown (filtered by those whose GO terms harbor fewer than 8 members). Full GO terms, p-values, fold enrichment and IDs are shown in the S5 Table. **B.** The top 100 dynamic IDs (ranked by number of dynamic nucleosomes per kbp) were searched for GO/REVIGO analysis annotation using TriTrypDB tools. The scatterplot shows a clusterization of GO terms (remaining after a redundancy reduction) in a two-dimensional semantic space, resulting in similar terms being plotted next to each other according to [75]. In general, more semantically similar GO terms are closer in the plot. **C.** Venn diagram of IDs with increased nucleosome dynamics (occupancy and/or fuzziness) in epimastigotes and TCTs.
(PDF)

**S9 Fig. Representative IGV snapshots of MNase-seq data mapped against the *T. cruzi* CL-Brener Esmeraldo-like genome at Chr 12S highlighting a CDS-containing region. Arrows indicate transcription direction.**
(PDF)

**S10 Fig. A.** Density of nucleosome occupancy (values on the y axis from Figs 4A and 5A–5H) for the indicated gene classes/features in epimastigotes (pink) and TCTs (green). The overlap percentage between life forms is indicated above each density plot. **B**. Box-plot of point by point distance (in y axis) from epimastigote and TCT landscapes (from Figs 4A and 5A–5H). A Tukey test was performed (considering intergenic regions as a control with few changes between epimastigotes and TCTs) to verify which gene classes/features have more differences between life forms. **C**. Representation of the results (95% confidence interval) of a pairwise Tukey test (intergenic as a control). Comparisons close to zero are considered not significantly different.
(PDF)

**S11 Fig. A**. Total length, in kbp, of members of the multigenic family. **B.** Number of dynamic and total (dynamic plus static) nucleosomes per kbp in multigenic family members, hypothetical genes and others (refers to all other IDs, including tRNA, rRNA, snRNA, snoRNA, mRNA, and pseudogene). **C.** Distribution of all nucleosomes, all dynamic nucleosomes, and dynamic nucleosomes in TCTs and epimastigotes (increase at occupancy and/or fuzziness) with regard to the distribution in disrupted and conserved *T. cruzi* genome compartments. DGF-1, GP63 and RHS are expected to be found in both compartments according to Berná et al. (2008). **D.** Nucleosomes per kbp. Conserved compartments have slightly more nucleosomes per kbp.
(PDF)

**S1 Table. Size of curated genomic regions.**
(XLSX)

**S2 Table. All dynamic nucleosomes with FDR < 0.05 for fuzziness and/or occupancy and/or position shift.**
(XLSX)

**S3 Table. Polycistrons with dynamic nucleosomes at dSSRs.**
(XLSX)

**S4 Table. GO terms enriched at polycistrons with at least 4 dynamic nucleosomes in their TSSs.**
(XLSX)

**S5 Table. GO terms enriched at intragenic regions.**
(XLSX)

**S6 Table. Data Requirements.** All numerical values that were used to generate graphs from Figs 1–7.
(XLSX)

## Acknowledgments

We thank Karin Cruz and Ivan Novaski Avino for technical assistance. We thank Herbert Silva for *in silico* evaluation of the proteomic levels of RNA Pol II components and for reading this manuscript and providing critical comments.

## Author Contributions

**Conceptualization:** Alex R. J. Lima, José Patané, M. Carolina Elias, Julia P. C. da Cunha.

**Data curation:** Alex R. J. Lima, Christiane B. de Araujo, Saloe Bispo, José Patané, M. Carolina Elias, Julia P. C. da Cunha.

**Formal analysis:** Alex R. J. Lima, Christiane B. de Araujo, Saloe Bispo, José Patané, Julia P. C. da Cunha.

**Funding acquisition:** Ariel M. Silber, M. Carolina Elias, Julia P. C. da Cunha.

**Investigation:** Alex R. J. Lima, Christiane B. de Araujo, Saloe Bispo, José Patané, M. Carolina Elias, Julia P. C. da Cunha.

**Methodology:** Alex R. J. Lima, Christiane B. de Araujo, Saloe Bispo, Julia P. C. da Cunha.

**Project administration:** Ariel M. Silber, M. Carolina Elias, Julia P. C. da Cunha.

**Resources:** Ariel M. Silber, M. Carolina Elias, Julia P. C. da Cunha.

**Software:** Alex R. J. Lima, Saloe Bispo, José Patané.

**Supervision:** José Patané, Ariel M. Silber, M. Carolina Elias, Julia P. C. da Cunha.

**Validation:** Alex R. J. Lima, Julia P. C. da Cunha.

**Writing – original draft:** Alex R. J. Lima, Julia P. C. da Cunha.

**Writing – review & editing:** Alex R. J. Lima, Christiane B. de Araujo, Saloe Bispo, José Patané, Ariel M. Silber, M. Carolina Elias, Julia P. C. da Cunha.

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
