## [Decision Letter · Decision Letter 0]

24 Nov 2020

Dear Dr da Cunha,

Thank you very much for submitting your manuscript "Nucleosome landscape reflects phenotypic differences in Trypanosoma cruzi life forms" for consideration at PLOS Pathogens. As with all papers reviewed by the journal, your manuscript was reviewed by members of the editorial board and by several independent reviewers. The reviewers appreciated the attention to an important topic. Based on the reviews, we are likely to accept this manuscript for publication, providing that you modify the manuscript according to the review recommendations.

The detailed comments are appended and most are very useful.

One short point from me (Christine Clayton): transcriptional down-regulation in non-dividing forms of kinetoplastids is not unprecedented. Trypanosoma brucei stumpy forms have almost no incorporation of 3H uridine into mRNA and also have very low levels of mRNA (E. Pays et al., Abrupt RNA changes precede the first cell division during the differentiation of Trypanosoma brucei bloodstream forms into procyclic forms in vitro. Mol. Biochem. Parasitol. 61, 107-114 (1993)). It is also impossible to induce expression from an RNA pol I promoter in the stumpy form (see e.g. B. Liu, K. Marucha, C. Clayton, The zinc finger proteins ZC3H20 and ZC3H21 stabilise mRNAs encoding membrane proteins and mitochondrial proteins in insect-form Trypanosoma brucei. Molecular microbiology 113, 430-451 (2020). Transcriptome results also suggest low activity in metacyclic forms, though no measurements of transcription have been made. I think you should refer at least to the Pays paper, since their results are consistent with your hypothesis for T. cruzi non-dividing forms.

Sincerely,

Christine Clayton

Associate Editor

PLOS Pathogens

Vern Carruthers

Section Editor

PLOS Pathogens

Kasturi Haldar

Editor-in-Chief

PLOS Pathogens

orcid.org/0000-0001-5065-158X

Michael Malim

Editor-in-Chief

PLOS Pathogens

orcid.org/0000-0002-7699-2064

Dear authors

Your paper has been read by three experts who all thought it was very interesting but had some suggestions for improvement. The detailed comments are appended and most are very useful.

One short point: transcriptional down-regulation in non-dividing forms of kinetoplastids is not unprecedented. Trypanosoma brucei stumpy forms have almost no incorporation of 3H uridine into mRNA and also have very low levels of mRNA (E. Pays et al., Abrupt RNA changes precede the first cell division during the differentiation of Trypanosoma brucei bloodstream forms into procyclic forms in vitro. Mol. Biochem. Parasitol. 61, 107-114 (1993)). It is also impossible to induce expression from an RNA pol I promoter in the stumpy form (see e.g. B. Liu, K. Marucha, C. Clayton, The zinc finger proteins ZC3H20 and ZC3H21 stabilise mRNAs encoding membrane proteins and mitochondrial proteins in insect-form Trypanosoma brucei. Molecular microbiology 113, 430-451 (2020). Transcriptome results also suggest low activity in metacyclic forms, though no measurements of transcription have been made. I think you should refer at least to the Pays paper, since their results are consistent with your hypothesis for T. cruzi non-dividing forms.

Reviewer Comments (if any, and for reference):

Reviewer's Responses to Questions

**Part I - Summary**

Reviewer #1: The authors present genome-wide nucleosome positioning maps in two life forms of Trypanosoma cruzi that display markedly different cell morphology, transcription and replication activity, and infectivity. Trypanosomes are an ancient eukaryotic branch that exhibit interesting genomic and epigenetic peculiarities, such as constitutive transcription of genes by RNA Pol II from long polycistronic gene clusters and an apparent absence of transcriptional regulation at the level of initiation. Similar genome-wide nucleosomal studies were published in related kinetoplastids L. major (Lombraña et al. 2016) and T. brucei (Maree et al. 2017, Wedel et al. 2017). Although some of the findings described here have been previously reported in these related organisms, the authors present novel data showing that gene classes associated with specific life forms are differentially enriched with dynamic nucleosomes. Integration of transcriptomic data revealed that these differential nucleosomal occupancies and positioning may likely be associated with important phenotypic differences in this organism. Interestingly, the authors show a correlation between nucleosome dynamics of specific gene classes and dSSRs, and RNA transcript levels and postulates that differential chromatin structures at these sites may influence RNA transcript levels.

Reviewer #2: Overall, this is a valuable study, which adds much-needed understanding to how gene expression is influenced by the organisation of the T. cruzi genome. The authors perform sequencing of MNase-digested DNA from replicating and non-replicating T. cruzi life cycle forms to evaluate nucleosome patterns across the parasite’s genome. Though this mapping approach has been described in the related trypanosomatids Leishmania and T. brucei, these data are valuable for two main reasons: (i) the T. cruzi genome is the least well understood of these trypanosomatids, due to the very large numbers of widely dispersed multigene families; and (ii) the paper asks, for the first time in trypanosomatids, if nucleosome organisation is altered in non-replicating cells, perhaps reflecting changes in gene expression. Despite this broad interest and importance, aspects of the paper could be improved.

Reviewer #3: This is a very well conducted and in-depth study that contributes to understanding the epigenetic regulation that accompanies life stage changes in Trypanosoma cruzi. The authors performed a genome-wide high-resolution nucleosome mapping comparing epimastigotes and trypomastigotes. Given the unique gene expression regulation mechanism in these parasites, mainly centered at the post-transcriptional level, the presented data is particularly significant. The researchers provided a thorough analysis of the nucleosomes landscape in both life stages and examined relevant biological issues. As stated by the authors, "results indicate that chromatin architecture, defined primarily by nucleosome positioning and occupancy, reflects the phenotypic differences found among T. cruzi life forms despite the lack of a canonical transcriptional control context." Therefore, this manuscript brings novel and relevant results and worthy of publication in PLoS Pathogens.

**Part II – Major Issues: Key Experiments Required for Acceptance**

Reviewer #1: None

Reviewer #2: The authors define (lines 100-102) the ‘strand switch regions’ (SSRs) that surround PTUs are being sites where transcription either converges (terminates, TTSs) or diverges (initiates, TSSs), and use this definition to locate SSRs in the genome based on CDS orientation (lines 177-179). Mapping of RNA polymerase II and histone variants in Leishmania and T. brucei tells us that there is another form of SSR, which is located within a directional gene cluster and where transcription of one PTU initiates and another PTU terminates (ie adjacent TSS and TTS sites). The authors’ approach ignores these SSRs, if they exist in T. cruzi. Two things need to be considered in this light. 1. Are there no histone variant ChIP data for T. cruzi that would allow these intra-directional gene cluster SSRs to be defined? If so, all data should be re-evaluated. 2. If such data is not available, the authors must explain this limitation in their approach (first results section, materials and methods), and should discuss in what ways the absence of being able to identify such SSRs might impact on the downstream data they present.

Through a number of pieces of analysis, the authors note that there is greater nucleosome occupancy at divergent SSRs in trypomastigotes compared with epimastogotes, and they suggest that this may be due to decreased transcription in the former rather than latter (e.g. ‘possibly reflecting less elongation of RNA Pol II through adjacent PTUs’, line 239). If this was true, it would represent control at the point of transcription initiation, affecting a whole PTU, which would be truly very interesting (indeed, an ‘unprecedented’ observation, line 248). However, is it true? The authors have used RNA derived from the two life cycle stages for other analysis, so they should map these data and ask if they observe changes in transcript levels across whole PTUs in the two life cycle, which would be predicted from their observation. If such an effect is seen, their arguments are strengthened; if not, they should explain why not and suggest alternatives for the nucleosome patterns.

Reviewer #3: (No Response)

**Part III – Minor Issues: Editorial and Data Presentation Modifications**

Reviewer #1: Corrections:

1) P11, L248-249: Sentence reads:

"These findings suggest an unpredict regulation maintained in T. cruzi chromatin reflecting phenotypic differences between replicative and nonreplicative forms."

It is unclear what the authors intend with this sentence and should be clarified.

2) Figure 1E and S6 shows the percentage of dynamic and static nucleosomes at different genomic features gene classes. It is unclear from the text and figure legends if these graphs are representative of epimastigotes, TCTs, or an average of the two life forms.

3) Figure 2D shows GO terms associated with dynamic nucleosomes. One GO term contains a typo, reading "regulation of mytotitiv cytokinesis", that should be corrected.

4) Figures 2D, 3A, and S8B show GO term enrichment. The y and x axis are labelled "semantic space" with scales. It is unclear what these labels and scales represent and should be either explained in the appropriate legends, or described more clearly in the figures to be easier to follow.

5) In the text, figure 3C is mentioned before 3A and 3B - authors should rearrange the figure panels so that data is presented in sequential order.

6) Figure 8A:

The legend for figure S8A reads: "...(filtered by those whose GO terms harbor f than 8 members)." This sentence should be clarified.

It continues "..IDs are shown in the Supplemental Table." Table S5 should be mentioned here.

The first panel of fig 8A reads "6980 static IDs", while the legend refers to "nondynamic IDs". Authors should use one of these terms consistently.

The second panel of fig 8A reads "3499 dynamics ID", one presumes this should be "dynamic", not the plural form.

7) Figure 4:

P14, L341 - 345: This sentence presumably refers to fig 4B, and should state it in the text.

P15 L347 - 349 discusses the nucleosomal landscape around TSSs and TTSs in the polycistronic environment and refers to fig 4B. From the fig 4 legend, it appears that this sentence should refer to fig 4 C and D, not B.

8) Fig 5:

Panels A - G should be included in legend

Panel 5H is not referred to in text and may be more appropriate in the supporting information section.

In addition, adding an IGV snapshot showing the data presented in 5H over a larger genomic region (100 kb) and indicating the PTU directionality (as in fig 2A) will provide a great overview of the NPO data. This will benefit the paper as well as the reader.

9) On the y-axis of fig S9C, "cSSR_center-intergenic_center" is partly cut off in figure.

10) P17 L399 - 401 reads: "In addition, when normalized by total length in kbp, DGF-1 genes had more dynamic nucleosomes per kbp than all other gene classes (Fig S10B)."

Fig S10B is labelled '# nuv dynamic per ID' and does not make sense in the context of what is being described. If the authors instead refer to the second panel of S10A, this statement makes more sense. From the legend it seems that the description of B fits to the second panel of A.

If the above is correct, the panel currently labelled B is not described in the legend or mentioned in the text. It appears that this panel is referred to in L405 - 406. The authors should rectify this.

11) P18 L423 reads: "...as shown in Fig 6D." This should be 6C.

12) P18 L434 indicates the authors classified the transcripts into 5 groups namely: highest, high, medium, low, and differential. However, figure 7 and its legend (P28 L688 - 693) only refers to 4 groups - high, medium, low, and differential, not highest. Authors should clarify this.

13) P26 L632 indicates that chromatin was digested with 1500U MNase. Authors should indicate whether this is in Sigma or Worthington units, as one unit Sigma MNase has the same activity as 85 Worthington units.

14) P27 L644 - authors should state the read length of paired-end reads obtained from Illumnina sequencing.

Reviewer #2: MNase-seq data is presented as IGV snapshots in three figures (Figs 1, 2 and 5), but in all cases it is impossible to fully evaluate these data. Most importantly, the authors need to provide values for the MNase-seq mapping in order that the extent of read depth can be seen, in order that we can understand how epimastigotes and trypomastigotes compare, and how large the differences (‘diff’) are. In addition, the other genomic features detailed, plus the meaning of dynamic and how his was assigned, needs explained to the reader; without this information, these figures are very hard to follow.

Lines 423-430. I’m afraid that I am not convinced by the authors’ suggestion that ‘MASP, TS and RHS genes have a similar landscape with a clear NDR near the first ATG followed by a well-positioned nucleosome downstream and a fuzziness pattern’, [which differs from] ‘DGF-1 and GP63 [which] are more similar to each other, with an NDR just upstream of the first ATG, followed by an increase in nucleosome occupancy all along the gene body’. Visually, I cannot see this difference between the 2 gene groups, which all appear very similar; to verify this, the authors need to more rigorously compare the MNase-seq profiles between the gene families. In addition, why are mucins not discussed, since the nucleosome pattern for this gene family looks different to all the others.

Reviewer #3: Page 8 - Line 84

What would be the meaning of: "most nucleosomes are poorly positioned."

Page 9 – line 101: . . . that function as TSSs and TTSs (transcription termination sites) [11, 12]. missing shouldn't it be: "transcription initiation and transcription termination sites, respectively"?

Page 9 – line 103: 5'end is a more precise term than 5'portion.

Page 9 – line 105: "gene expression is remarkable," shouldn't it be something similar to "the control of gene expression which is remarkable."

Page 9 – line 110: Instead of "A second interesting biological feature," wouldn't it be better to use "Another interesting biological…"?

Page 17 – line 307: there is an incorrection "two DNA topoisomerases from classes II and II."

Page 17 – paragraph starting on line 306: A general comment regards the citation of genes by their TriTryp DB identity. It turns out that reading a text with these TriTryp identities is not agreeable and nor informative. Also, in the same paragraph: what is a "generic ribosomal protein"?

Page 26 – lines 520-524: The sentence starting on "In silico analysis of RNA Pol II component expression from public . . ." is confusing. More importantly, if the authors are not inclined to present their data, they should eliminate this sentence.

PLOS authors have the option to publish the peer review history of their article (what does this mean?). If published, this will include your full peer review and any attached files.

Reviewer #1: **Yes: **Hugh -G Patterton

Reviewer #2: No

Reviewer #3: No
---

## [Editor Report · Decision Letter 1]

28 Dec 2020

Dear Dr da Cunha,

Thank you very much for submitting your manuscript "Nucleosome landscape reflects phenotypic differences in Trypanosoma cruzi life forms" for consideration at PLOS Pathogens. As with all papers reviewed by the journal, your manuscript was reviewed by members of the editorial board and by several independent reviewers. The reviewers appreciated the attention to an important topic. Based on the reviews, we are likely to accept this manuscript for publication, providing that you modify the manuscript according to the review recommendations.

Thanks you for submitting the revised version. You seem to have addressed all of the issues raised. Please however, to save possible problems at the proof stage, could you make some corrections to the new text, as follows:

"Once gene expression is mainly regulated by posttranscriptional mechanisms" should be "Since gene expression...

The phrase "an unpredict regulation" doesn't make sense. Do you mean "unexpected regulation"? (Actually I'm not even sure "unpredict" exists, but if it does the meaning isn't appropriate.)

"Mucins show a nucleosome landscape that look different" should be "looks different".

"the steady-state levels of transcripts is the result of the balance among transcription expression, expression, RNA degradation and stabilization" should be

"are the combined result of transcription, RNA processing, and mRNA degradation". (Stablization affects degradation so isn't a separate contributor; and processing has to be mentioned.)

Line 504: delete: "as soon as they were identified in T. cruzi". Alternatively you could say: "since such sites have not yet been mapped in T. cruzi".

Sincerely,

Christine Clayton

Associate Editor

PLOS Pathogens

Vern Carruthers

Section Editor

PLOS Pathogens

Kasturi Haldar

Editor-in-Chief

PLOS Pathogens

orcid.org/0000-0001-5065-158X

Michael Malim

Editor-in-Chief

PLOS Pathogens

orcid.org/0000-0002-7699-2064

Thanks you for submitting the revised version. You seem to have addressed all of the issues raised. Please however, to save possible problems at the proof stage, could you make some corrections to the new text, as follows:

"Once gene expression is mainly regulated by posttranscriptional mechanisms" should be "Since gene expression...

The phrase "an unpredict regulation" doesn't make sense. Do you mean "unexpected regulation"? (Actually I'm not even sure "unpredict" exists, but if it does the meaning isn't appropriate.)

"Mucins show a nucleosome landscape that look different" should be "looks different".

"the steady-state levels of transcripts is the result of the balance among transcription expression, expression, RNA degradation and stabilization" should be

"are the combined result of transcription, RNA processing, and mRNA degradation". (Stablization affects degradation so isn't a separate contributor; and processing has to be mentioned.)

Line 504: delete: "as soon as they were identified in T. cruzi". Alternatively you could say: "since such sites have not yet been mapped in T. cruzi".
---

## [Editor Report · Decision Letter 2]

4 Jan 2021

Dear Dr da Cunha,

We are pleased to inform you that your manuscript 'Nucleosome landscape reflects phenotypic differences in Trypanosoma cruzi life forms' has been provisionally accepted for publication in PLOS Pathogens.

Best regards,

Christine Clayton

Associate Editor

PLOS Pathogens

Vern Carruthers

Section Editor

PLOS Pathogens

Kasturi Haldar

Editor-in-Chief

PLOS Pathogens

orcid.org/0000-0001-5065-158X

Michael Malim

Editor-in-Chief

PLOS Pathogens

orcid.org/0000-0002-7699-2064

Thank you for making the additional minor changes.
---

## [Editor Report · Acceptance letter]

19 Jan 2021

Dear Dr da Cunha,

We are delighted to inform you that your manuscript, " Nucleosome landscape reflects phenotypic differences in Trypanosoma cruzi life forms ," has been formally accepted for publication in PLOS Pathogens.

Best regards,

Kasturi Haldar

Editor-in-Chief

PLOS Pathogens

orcid.org/0000-0001-5065-158X

Michael Malim

Editor-in-Chief

PLOS Pathogens

orcid.org/0000-0002-7699-2064